# Angiogenesis Inhibitors for Colorectal Cancer. A Review of the Clinical Data

**DOI:** 10.3390/cancers13051031

**Published:** 2021-03-01

**Authors:** Torben Frøstrup Hansen, Camilla Qvortrup, Per Pfeiffer

**Affiliations:** 1Danish Colorectal Cancer Center South, Vejle University Hospital, 7100 Vejle, Denmark; 2Institute of Regional Health Research, University of Southern Denmark, 5000 Odense, Denmark; 3Department of Oncology, Rigshospitalet, 2100 Copenhagen, Denmark; camilla.qvortrup@regionh.dk; 4Department of Oncology, Odense University Hospital, 5000 Odense, Denmark; per.pfeiffer@rsyd.dk; 5Department of Clinical Research, University of Southern Denmark, 5000 Odense, Denmark

**Keywords:** Angiogenesis, colorectal cancer, monoclonal antibodies, small molecule tyrosine kinase inhibitors

## Abstract

**Simple Summary:**

Targeting angiogenesis, the formation of new blood vessels, is an integral part of many cancer treatments, including colorectal cancer. The overall clinical benefit is well documented but modest. It has been an ongoing task for the last decade to isolate patient and tumor characteristics instrumental in identifying the subgroups to truly benefit; so far with limited success. The introduction of immunotherapy has opened a new era for anti-angiogenic treatment, as these two therapeutic strategies seem to work in synergy. This review will highlight the clinical achievements of anti-angiogenic treatment of colorectal cancer since 2004 and elaborate on the perspectives of combining it with immunotherapy.

**Abstract:**

Since the late 1990s, therapy for metastatic colorectal cancer (mCRC) has changed considerably, and the combination of doublet or triplet chemotherapy and a targeted agent are now routinely used. The targeting of angiogenesis, the development of new blood vessels, represents a key element in the overall treatment strategy. Since the approval in 2004 of the first anti-angiogenetic drug, multiple agents have been approved and others are currently under investigation. We present an overview of the recent literature on approved systemic treatment of mCRC, with a focus on anti-angiogenic drugs, and current treatment approaches, and elaborate on the future role of angiogenesis in colorectal cancer as seen from a clinical perspective. The treatment of mCRC, in general, has changed from “one strategy fits all” to a more personalized approach. This is, however, not entirely the case for anti-angiogenetic treatments, partly due to a lack of validated biomarkers. The anti-angiogenetic standard treatment at the present primarily includes monoclonal antibodies. The therapeutic field of angiogenesis, however, has received increased interest after the introduction of newer combinations. These approaches will likely change the current treatment strategy, once again, to the overall benefit of patients.

## 1. Introduction

### 1.1. Colorectal Cancer

Worldwide, 1.8 million new patients are diagnosed each year with colorectal cancer (CRC). Approximately half of the patients will be diagnosed with metastatic CRC (mCRC), at either the time of diagnosis (synchronous) or due to later recurrence (metachronous) [1]. Almost half the number of new cases, 0.86 million, die each year.

### 1.2. Treatment Overview

For several years, the armamentarium of standard treatment for patients with mCRC have included combination chemotherapy with either 5-FU, oxaliplatin, irinotecan, or two classes of targeted therapies [2,3]. These therapies inhibit the signaling pathways related to the epidermal growth factor (EGF) and vascular endothelial growth factor (VEGF) receptors. The monoclonal antibodies cetuximab and panitumumab, targeting the EGF receptor (EGFR) and bevacizumab, targeting the VEGF-A ligand, are the most commonly used in the field of mCRC.

It is well known that the benefit of anti-EGFR is restricted to around 40% of the patients who are *RAS* and *BRAF* wild type (wt) [2,3]. The common treatment approach has changed from single agent chemotherapy to a doublet regimen, or occasionally triplet chemotherapy regimen, often in combination with bevacizumab, cetuximab, or panitumumab based on the *RAS* mutational status. Typically, anti-EGFR therapy improves major efficacy parameters (response rate, PFS and OS) when added to doublet regimens like 5-FU + irinotecan (FOLFIRI) or 5-FU + oxaliplatin (FOLFOX) but results were more equivocal when bevacizumab was added to modern infusional doublet regimens (Table 1). Nerveless, the optimal combination of chemotherapy and targeted therapy for first line therapy have been debated for many years. Three randomized trials have directly compared efficacy of EGFR inhibitors and bevacizumab in patients with RASwt mCRC, but with a very heterogeneous picture and no well-founded conclusion. Prior studies have shown that left-sided mCRC are dependent on EGFR related pathways and when investigators from the major cooperative groups pooled data in patients with left-sided tumors, it became evident, and the efficacy data became much more homogenous, showing a clear advantage of EGFR inhibitors with higher overall response rates (ORR) and prolonged overall survival (OS) in patients with left-sided primaries [4,5]. However, there is currently no solid evidence indicating that *RAS* mutations should render anti-VEGF-A therapy obsolete in the setting of mCRC.

A number of randomized studies, pioneered by Dr. Falcone’s group, have evaluated triplet chemotherapy (FOLFOXIRI) in patients unselected by the RAS mutational status. The FOLFOXIRI regimen does have a significant toxicity-profile, thus requiring patients to exhibit a good performance status. Consequently, patients included in the FOLFOXIRI trials are more often younger or more often in performance status 0 than usually in clinical trials. Two Italian phase III trials [15,16] showed that triplet chemotherapy was more effective than a doublet (either FOLFIRI or FOLFOX) in terms of ORR, PFS and OS. In the TRIBE–study, bevacizumab was added to both the triplet and the doublet combinations, and thus we can only conclude that a triplet chemotherapy can be safely combined with bevacizumab but whether bevacizumab adds to the efficacy of a triplet cannot be concluded from these studies [16]. In the randomized phase II OLIVIA trial [17], in which mCRC patients with liver-limited disease were included, a triplet chemotherapy with bevacizumab produced a very impressing ORR of 81%. Consistently, a high response rate of at least 60% was observed in all studies that evaluated FOLFOXIRI with or without bevacizumab [3].

### 1.3. Angiogenesis

Basically, the term vasculogenesis describes the process of the initial endothelial differentiation of angioblasts during embryogenesis [18], whereas angiogenesis refers to the formation of new blood vessels from existing endothelial cells [19]. The regulation of the angiogenic process is a complex balance between stimulating and inhibiting stimuli. The VEGF system consists of six ligands and three receptors (VEGFR). The VEGF-A ligand is the most important. It is secreted by multiple cell types including the malignant cells and stimulates endothelial cell (EC) differentiation, migration, growth and survival [20]. The receptor that is primarily responsible for transmitting this VEGF-A-mediated signal in the EC is VEGFR-2, whereas the role of VEGFR-1 probably is more regulatory and inhibitory [21]. The autonomic growth pattern that characterizes malignant neoplasms are contributing to the fact that malignant tumors are often hypoxic to varying degrees. This hypoxia leads to increased transcription of a large number of genes, including *VEGF-A*, with the common purpose of ensuring a more adequate oxygenation of the tumor [22]. It is the so-called hypoxia-inducible factors (HIF) that are the cause of this gene regulation. The three members are formed from the oxygen-sensitive subunits (HIF-1α, HIF-2α and HIF-3α) and the non-oxygen-sensitive HIF-1β subunit [23]. During hypoxia, HIF-1α (the best described) stabilizes and translocate to the cell nucleus to form the activated HIF-1 complex together with HIF-1β.

### 1.4. Anti-angiogenetics 

The therapeutics targeting angiogenesis are divided into two main groups: the monoclonal antibodies (mAbs) and the small molecules, tyrosine kinase inhibitors (TKIs). The mAbs exert their action by either directly binding VEGF-A or blocking the extracellular binding domain of the corresponding receptor. Bevacizumab (Avastin^®^) binds to all isoforms of VEGF-A and aflibercept (Zaltrap^®^) a soluble decoy receptor binds VEGF, thereby preventing activation of their endogenous receptors whereas ramucirumab (Cyramza^®^) binds with high affinity to the VEGFR-2 extracellular domain, which prevents binding of VEGF ligands and thereby inhibiting receptor activation. The TKIs exert their anti-angiogenetic effect after internalization in the cell and binding to, and inhibiting, the kinase domain of the various receptors involved in the angiogenetic process (tyrosine kinase, serine/threonine kinase or dual protein kinase inhibitors).

### 1.5. Current Challenge 

Inhibition of tumor-associated angiogenesis have been utilized for the treatment of patients with mCRC for more than 15 years [2,3]. Since the initial approval of bevacizumab in 2004, several other agents have been investigated within phase III trials, leading to several additional approvals. 

The obtained survival benefit from these drugs is often limited due to multiple resistance mechanisms and all attempts to individualize treatment have so far been unsuccessful. This class of therapeutics are consequently administered to a broad and unselected patient population constituting a social-economic challenge to the community. The adverse events related to these treatments, although often manageable, may sometimes be severe and even fatal. This scenario calls for the identification of predictive biomarkers or new treatments combinations with a more favorable advantage/disadvantage ratio if this field of therapy is to evolve even further.

## 2. Existing Treatment

Presently, targeting angiogenesis for the treatment of mCRC may be applied to all treatment lines. Bevacizumab is used in combination with chemotherapy in both first and later lines of therapy, ramucirumab and aflibercept, together with chemotherapy, is used in the second line setting and finally regorafenib given as monotherapy is used for patients with chemo-refractory disease. In brief, addition of an antibody targeting VEGF or VEGFR (such a bevacizumab, aflibercept or ramucirumab) to second line treatment significantly improved OS by a median of 1.4 to 2 months in all second line trials, independently whether a VEGF inhibitor had been used before. In total, four second line trials have reported a gain in OS by the addition of an antiangiogenic compound, irrespective of the various first-line regimens [2,3].

### 2.1. Bevacizumab in First Line

The most widely used vascular inhibitor is bevacizumab. Bevacizumab as monotherapy has no or only very modest effect in mCRC and is most often used in combination with chemotherapy. The first randomized trials demonstrated that bevacizumab improves the efficacy of chemotherapy as measured by three key efficacy parameters (response rate (RR), progression-free survival (PFS) and OS). In combination with IFL, a bolus regimen consisting of irinotecan and 5-fluorouracil (5FU), OS was extended by almost five months to 20 months [7]. As there were only a few additional side effects at the same time, bevacizumab with combination chemotherapy quickly became standard in most parts of the world, and since then bevacizumab has been on the top list of the best-selling drugs, presently with annual sales of about 7 billion USD [24]. However, due to inferior efficacy, IFL was subsequently substituted by modern combination regimens (e.g., CapOx (capecitabine and oxaliplatin), FOLFOX (5FU and oxaliplatin) or FOLFIRI (5FU and irinotecan)). When bevacizumab was tested in combination with CapOx or FOLFOX, gain in efficacy was lower than expected [13]. The PFS was prolonged by a modest 1.4 months, and surprisingly, no significant improvement in the confirmed RR (38% vs. 38%) or OS (19.9 vs. 21.3 months) was seen. In Table 1, an overview of the principal clinical trials addressing bevacizumab in the first line treatment of mCRC is provided [6,7,8,9,10,11,12,13,14]. Briefly, trials testing monotherapy or bolus combination chemotherapy regimens with bevacizumab showed improvement in all efficacy parameters, but this is somehow in contrast to the NO16966 and the ITACa trials, where response rates and OS were not upgraded.

### 2.2. Bevacizumab in Later Lines

A number of well-conducted randomized trials have documented the efficacy of bevacizumab when combined with 5FU as monotherapy [25] or when combined with chemotherapy after first-line treatment (even if bevacizumab was already part of first-line treatment) (Table 2) [26,27,28,29,30,31,32,33,34].

A: 30% had received bevacizumab as part of first line therapy. The benefits of aflibercept with FOLFIRI were observed in subgroups of patients with or without prior bevacizumab treatment. B: BEBYP was interrupted prematurely after accrual of 184/262 planned patients. Thus, there is no doubt that bevacizumab has clinically significant activity, but the challenge in modern oncology is to choose the right treatment for the right patients [2,3]. Unfortunately, to date, there are no approved or generally accepted biomarkers for predicting benefit from bevacizumab. This is in contrast to one of the other very frequently used treatment principles in mCRC, namely the targeting of EGFR with monoclonal antibodies (panitumumab or cetuximab), in which mutation status of *RAS* in the MAPK pathway has been proven of predictive value [2,3]. The value of *RAS* mutational status in angiogenesis inhibition in CRC, on the other hand, has been more unclear. For the past 15 years, bevacizumab has been claimed to exert its effect independently of *RAS* status, but this has never been studied regularly, and some subgroup studies suggest that its effectiveness may be limited in patients with *RAS*-mutated tumors [3].

Due to the knowledge on both the advantage of anti-EGFR and anti-angiogenic therapy and supported by promising preclinical data, it was obvious to test if multi-blockade with the combination of anti-angiogenic and anti-EGFR therapy could improve survival even further. Initial clinical data supported the hypothesis as a randomized phase II study [35] showed that the combination of irinotecan, cetuximab and bevazicumab resulted in a higher RR and longer PFS than cetuximab and bevazicumab in patients with pre-treated mCRC. However, despite the above-mentioned promising results on double-blockade in preclinical models, and from early clinical data, two large phase III studies—the CAIRO2 and the PACCE studies—failed to confirm these results (Table 3) [36,37,38,39,40]. Both trials showed that addition of bevazicumab to an anti-EGFR antibody and combination chemotherapy in chemo-naïve patients was associated with an inferior outcome compared to an anti-EGFR antibody and combination chemotherapy alone [36,37].

Since tumors cannot grow to more than 2–3 mm^3^ without blood supply, it was also obvious to investigate the effect of bevacizumab in the adjuvant setting. Unfortunately, in two large randomized trials, no gain was measured in terms of OS in patients with CRC, and in one study, there was fewer patients alive after ten years than in the control group [41,42]. It is not entirely clear why anti-VEGF-A therapy was ineffective in the adjuvant setting. This may be related to the fact that adjuvant treatment often targets microscopic clusters of cells or even single cells in the circulation situations where the tumor-related blood vessels may not be dependent on VEGF-A to the same extent as in the metastatic setting.

Bevacizumab is in general well tolerated, however vascular-related side effects have been seen with the most serious being gastrointestinal perforation, hemorrhage and arterial thrombosis (in less than 1% of patients). More commonly proteinuria and hypertension. In a meta-analysis by Zhu et al. grade three hypertension was reported in approximately 9% of patients treated with low-dose bevazicumab and in 16% of patients receiving doses of 10 mg/kg or above [43].

### 2.3. Ramucirumab

In combination with FOLFIRI, second line therapy with ramucirumab did not increase RR but significantly prolonged PFS from 4.5 to 5.7 months and OS from 11.7 to 13.3 months following first-line treatment with fluoropyrimidine, oxaliplatin and bevacizumab [31].

### 2.4. Aflibercept

The anti-angiogenic fusion protein aflibercept also produce a survival advantage when added to FOLFIRI in patients progressing on a prior oxaliplatin-containing regimen [28]. The RR was increased from 11 to 20%, PFS was prolonged from 4.7 to 6.9 months and OS from 12.1 to 13.5 months and the benefit was observed independent of prior bevacizumab.

### 2.5. Tyrosine Kinase Inhibitors

The other main group of anti-angiogenic drugs—the TKIs have also been tested in the mCRC population. Due to the targeting of multiple signalling pathways beyond the VEGFR full dose TKI may be difficult to tolerate and often requires dose modifications, and are most often administered as monotherapy.

More than 10,000 patients have been included in randomized phase II and III trials investigating the TKIs. Many randomized trials were initiated soon after the turn of the century, sometimes even without solid phase II data [44]. In broad terms, first- and second-line trials (Table 4) [45,46,47,48,49,50,51,52,53,54] tested the TKIs in combination with chemotherapy whereas TKI monotherapy was evaluated in late line trials (Table 5) [55,56,57,58,59,60,61,62,63]. In general, first line trials aimed to prolong PFS with 2–3 months and later line trials to prolong OS with 2 months. Several different TKIs were evaluated in large, well-conducted trials including more than 1000 patients, but in general with disappointing results. A total of 2000 patients participated in two large randomized trials comparing FU monotherapy with or without semaxinib or comparing combination chemotherapy with or without semaxinib, but unfortunately results from these trials have never been published.

In the HORIZON II first line trial, cediranib prolonged PFS (secondary endpoint) significantly from 8.3 to 8.6 months but without any prolongation of OS [46]. In the CONFIRM II second line trial, vatalanib prolonged PFS (secondary endpoint) significantly from 4.2 to 5.6 months but without any prolongation of OS [51]. Apart from these two trials showing a modest prolongation of PFS no other trials have shown that TKI add to the efficacy of chemotherapy by extending PFS and no randomized trial has shown an OS benefit neither in first line, nor in second line. 

In contrast to these depressing results of TKI with chemotherapy, monotherapy TKI compared to placebo has demonstrated a significant benefit in several efficacy parameters and for regorafenib the advantage was proven in several comparable trials. In the largest trial, CORRECT, with 800 chemo-refractory patients with mCRC, PFS significantly was prolonged from 1.7 to 1.9 months (HR 0.49) and OS from 5.0 to 6.4 months (HR 0.77). 

Thus, despite a large number of well-conducted clinical trials, regorafenib still remains the only TKI occasionally used in the clinical practice of mCRC (Table 4 and Table 5). The most frequent adverse reactions (ARs) in patients receiving regorafenib is fatigue, rash or hand-foot skin reaction, diarrhea, and anorexia and often dose-reductions are required to handle regorafenib-related adverse reactions. Several trials have shown that a lower starting dose with gradual dose-escalation is an alternative, safe and better tolerated approach for administration of regorafenib and this strategy should be preferred in clinical practice [57,64].

### 2.6. Economy

Anti-angiogenetic therapy is used in an unselected manner, in line with the standard chemotherapy approved for mCRC, due to the lack of validated predictive biomarkers. One consequence is a very broad application, and most patients with mCRC are exposed, at least once, to this class of therapy. As the overall benefit from this addition often is rather limited, and the treatment is very expensive, this has naturally triggered speculations as to the cost-effectiveness of this approach. 

This theme is addressed in many papers. The conclusions may differ slightly depending on prices in the individual countries, and differences in the willingness-to-pay value for a given outcome, but overall, addition of anti-angiogenetic treatment (often bevacizumab in these calculations) to palliative chemotherapy in mCRC is not cost-effective under the current circumstances. This was also the conclusion in a publication from 2017, by Goldstein et al., summarizing that the addition of bevacizumab to first-line chemotherapy in mCRC failed to be cost-effective in five different countries [65]. The highest incremental cost-effectiveness ratio was demonstrated for the U.S. with 571,000 USD per quality-adjusted life years, more than three times higher than the willingness to pay threshold. Similar conclusions have been obtained for bevacizumab used as maintenance therapy in combination with capecitabine and as a regular second line treatment as well.

With a possible expansion of the indication for the use of anti-angiogenetic treatment, considering the potential benefit combining these therapies to immunotherapy, these economic considerations will once again be highly relevant as both cost and benefit will likely change. 

### 2.7. Biosimilars

Presently biosimilars of bevacizumab are under investigation in different clinical trials including randomized studies comparing original bevacizumab with chemotherapy and biosimilars with chemotherapy and in near future the results are expected. Patent and regulatory exclusivities will protect Avastin^®^ until at least June 2020, but maybe until January 2022. Two biosimilar to bevacizumab have been approved for use in the European Union, (Mvasi^®^ by Amgen and Zirabev^®^ by Pfizer) in 2018 and 2019, respectively, but marketing of these two biosimilars has been delayed until relevant regulatory exclusivities have expired [66]. 

### 2.8. Resistance Mechanisms

Like other cancer treatments, resistance to drugs targeting the angiogenetic process will lead to disease progression. Resistance is a completely natural consequence due to the incredibly complex regulation controlling the angiogenic process. It is difficult to differentiate between resistances to anti-angiogenic therapy in certain scenarios, such as CRC, where these drugs are used in combination with chemotherapy. At present, we have only limited insight into anti-angiogenic resistance mechanisms, but they can generally be divided into mechanisms that are due to pre-existing conditions or have been acquired due to the treatment. Examples of the former are heterogeneity in the tumor-associated blood vessels (some are immature and vulnerable to antiangiogenic treatment, while others are not), organ and tumor-specific differences in the regulation of angiogenesis, bioavailability of the drug also known from resistance to chemotherapy (drug transport, tumor architecture, vascular delivery), genetic differences between individuals that may explain differentiated responses to treatment, differences in which factors primarily drive angiogenesis in the primary tumor and metastases, the specific mono-targeting of ECs leaving supportive structures such as basement membrane and pericytes for rapid regrowth [67,68]. Acquired conditions are known for upregulation of antiapoptotic and alternative proangiogenic factors that are upregulated as a consequence to a single target inhibition as seen with anti-VEGF-A, selection of hypoxia-resistant tumor cells, alternative vascularization, co-option, and increased tumor aggressiveness or epigenetic upregulation of antiapoptotic factors in the target cells [69,70,71].

## 3. Scientific Rationale

### 3.1. Introduction

The overall rationale for targeting the process of angiogenesis is the consequence of decades (centuries) of basic and clinical research. Angiogenesis is involved in several physiological processes including wound healing and menstrual cycle, but angiogenesis may as well be involved in pathophysiological conditions characterized by either insufficient or excessive blood vessel formation as seen in malignant neoplasms. Angiogenesis plays a critical role in the continued growth of cancer because solid tumors need a blood supply if they are to grow beyond a few millimeters. Tumor-associated blood vessels are structurally and functionally abnormal and characterized by an irregular chaotic network of leaky blood vessels, resulting in elevated intratumoral pressure [72]. In 1971, Judah Folkman described tumors’ dependence on newly formed blood vessels waking the interest in angiogenesis as a process for pharmaceutical targeting [73,74]. The factor, primarily responsible for stimulating the formation of new blood vessels (VEGF-A) was identified in 1989 by Napoleone Ferrara [75].

### 3.2. Basic Tumor-associated Angiogenesis

Tumor growth beyond 2 mm^3^ is the main initiating event in tumor-associated angiogenesis. At this stage simple diffusion is no longer sufficient and the cancer cells with the longest distance to the existing blood vessels become hypoxic. This triggers the secretion of VEGF-A from the hypoxic cells that eventually interact with ECs (through VEGFR-2) in nearby blood vessels. This shifts the affected ECs from a dormant state to an active proliferative state, classically described as the angiogenic switch [76]. The initiated cascade of events constitutes the degradation of the blood vessel integrity, cleavage of the extracellular matrix, vasodilation, increased permeability, migration of ECs and the forming of so-called sprouts. Once connected to adjacent sprouts the entire process reverses in order to stabilize and mature the newly formed blood vessel [77]. It is during this maturation process that VEGF-A acts as a crucial survival factor [78].

### 3.3. Normalization

The structural and functional abnormalities of tumor-associated blood vessels, as previously discussed, provides several advantages for a growing tumor. It ensures nutrients and oxygen, it provides the basic escape route for potential metastatic cancer cells, and it impairs the extravasation of larger molecules such as chemotherapy and thus some degree of treatment resistance. Reversing these processes through anti-angiogenetic treatment are summarized in the normalization theory presented by Rakesh Jain [72] and later proven clinically. It has recently been argued that the optimal window for normalization is rather narrow and over pruning may impair delivery of chemotherapy just as much as insufficient targeting of the tumor vasculature.

## 4. Anti-angiogenesis and Immunotherapy

The limited benefit from anti-angiogenetic therapies in mCRC have paved the way for new combinations in order to sustain tumor control. The currently most promising approach is the combination with immunotherapy.

### 4.1. Tumor-Microenvironment

Tumor-associated blood vessels have a unique architecture and physiologic properties as previously addressed. These properties lead to the generation of an immune suppressive tumor microenvironment [79]. Malignant tumors have the ability to evade immune suppression through changes in the recruitment, trafficking and infiltration of effector T cells and their final recognition and killing of cancer cells. This is the consequence of several processes related to tumor-initiated angiogenesis. The initial up-regulation of VEGF-A inhibit the maturation of dendritic cells (DCs) [80], a crucial and initial step in the process of immunity, and lead to an upregulation of programmed death-ligand 1 PD–L1 on DCs further suppressing T cell function [81]. Leaky blood vessels increase the intra-tumoral pressure and together with the down-regulation of cell-adhesion molecules complicates extravasation of tumor infiltrating lymphocytes (TILs). Hypoxia itself lead to upregulation of PD–L1 and the simultaneous up-regulation of VEGF-A which furthermore impairs the function of the antigen presenting cells. Finally, the balance of TILs shifts towards increased infiltration of regulatory T cells (Treg) on behalf of cytotoxic effector cells (CD8^+^) due to regulatory changes in the ECs [82]. This is primarily due to an increased expression of FAS ligand on tumor-associated ECs that prevents effector T cells from crossing the EC barrier by inducing apoptosis. Treg are resistant to FAS ligand creating a relative overexpression of Treg in the tumor compared to the effector cells. Treg furthermore inhibit the antigen presenting cells in the tumor enhancing the immune suppressive environment. Tumor vasculature normalization thus creates an immune friendly microenvironment and may turn a “cold” tumor into a “warm”. Added to this is the recent discovery of how stimulated immune cells themselves lead to vascular normalization, partly through CD8^+^ T–cells and interferon gamma (IFN–Ƴ), creating a beneficial immune-vasculature crosstalk and a rationale for combining these two classes of therapeutics with a potential synergistic benefit [83]. The optimal dosing and timing of these treatment combinations will likely differ between individual tumor types and represent an essential key for unlocking their full potential.

### 4.2. Anti-angiogenetic Therapy and Immunotherapy in Colorectal Cancer

This potential synergism between anti-angiogenic and immune checkpoint inhibitor drugs has resulted in numerous clinical trials testing the combination of PD–1/PD–L1 antibodies with anti–VEGF drugs. By now, a number of randomized trials have shown remarkable results which have led to approvals by the FDA and/or EMA in renal cell carcinoma (axitinib plus pembrolizumab, and axitinib plus avelumab), endometrial carcinoma (pembrolizumab plus lenvatinib), non-squamous NSCLC (bevacizumab and atezolizumab), and hepatocellular carcinoma (bevacizumab plus atezolizumab). This broad clinical activity suggest that a combination strategy may also be of benefit in colorectal cancer [84].

The first clinical data in CRC were presented by Bendell et al. at the ASCO–GI conference in 2015 [85]. Among 13 patients with treatment refractory disease, they demonstrated one objective tumor response by adding bevacizumab to MPDL3280A (atezolizumab). The following year, at the AACR 107th annual meeting, Wallin et al. presented the results from 23 patients with mCRC treated with first-line FOLFOX, bevacizumab, and atezolizumab [86]. They demonstrated promising efficacy data, with a median PFS of 14.1 months and parallel translational research argued for immune-related activity by this combination. The corresponding papers to these two initial abstracts are so far not published. 

In 2017, Yoshida et al. published the results of their pilot study (COMVI study) in anticancer research [87]. Six patients with previously untreated mCRC were included in a prospective single arm study. All patients received, as standard therapy, oxaliplatin (130 mg/m^2^) on day one, capecitabine (1000 mg/m^2^) twice daily on the days 1–14, and bevacizumab (7.5 mg/kg) on day one. To this backbone, they added cultured αβ T–lymphocytes (>5 × 10^9^) combined with interleukin–2 and anti–CD3 on day 18. Two patients achieved a complete response, three a partial response, and one demonstrated stable disease as the best outcome. The median progression free and overall survival was 567 and 966 days, respectively. Adverse events were mild to moderate. Although a small pilot study, these published results, for the first time in patients with CRC, demonstrated that combining chemotherapy and anti-angiogenesis with immune-modulating therapies were feasible and efficacy data were promising.

An additional two abstracts were presented the following years but they did not quite meet the initial expectations. In 2018, at the ESMO congress, Grothey et al. presented a late-breaking abstract from cohort 2 of the MODUL trial [88]. After induction therapy with the FOLFOX + bevacizumab regimen, 445 patients were randomized to maintenance fluoropyrimidine and bevacizumab ± atezolizumab. The updated analyses revealed no difference in median PFS and OS between the two strategies. Mettu et al. presented an abstract at the poster discussion session at the same congress the following year [89]. In this study, 133 patients with mCRC, with treatment resistant disease, were randomized to receive capecitabine and bevacizumab plus placebo or atezolizumab as last line treatment. The study reached its primary endpoint demonstrating a significant improvement of PFS by the addition of atezolizumab, although the numerical difference was only one month. The corresponding manuscripts have not been published either. 

The first publication based on a commercially available immuno-therapeutic, within this specific field in CRC, was published earlier this year, in April 2020, and revealed the results from the dose expansion phase Ib trial REGONIVO (EPOC1603) [90]. Fukuoka et al. included patients with gastric or CRC, 25 of each, who had progressed on a minimum of two previous lines of palliative treatment. All the patients with CRC had previously received anti-angiogenetic treatment, the cancer in one patient had deficient mismatch repair (dMMR) but the remaining 24 were all proficient (microsatellite stable), and six had *RAS* mutations. Patients were treated with nivolumab 3 mg/kg every two weeks and regorafenib once daily, day 1–21, in a four-week cycle. During the dosing-finding, part of the study regorafenib was reduced from initially 160 mg to the recommended 80 mg at which no patients experienced dose-limiting toxicity. Among the patients with CRC 9 (36%) achieved an objective tumor response and median PFS was 7.9 months. A trend towards better outcome for the patients with lung metastases, compared to liver metastases was presented, which could be due to a more immunosuppressive environment in the liver compared to the lung. This study provided real clinical evidence of synergy between the investigated drugs. Neither of the two would be expected to provide meaningful benefit as singe agents in this group (except the one patient with a dMMR tumor) but combined, one out of three responded. 

These results, together with similar findings in other cancer types, have paved the way for multiple trials assessing the clinical benefit from combining immunotherapy and anti-angiogenetic treatment in CRC. An example is the recently published study protocol AtezoTRIBE by Antoniotti et al. [91]. In this randomized phase II trial untreated patients with unresectable mCRC will receive FOLFOXIRI and bevacizumab ± atezolizumab in four months followed by maintenance 5–fluoruracil, leucovorin, bevacizumab ± atezolizumab. The study is estimated to be completed in April 2021. A supplementary search at clinical.trials.gov for trials in CRC combining immunotherapy with an anti-angiogenetic drug revealed more than 20 ongoing clinical trials (Table 6). With a specific focus on CRC only, this level of clinical activity underlines the potential impact gained by combining these two classes of therapeutics.

## 5. Conclusions

For many years, the ability to suppress angiogenesis has been exploited in the field of oncology. The efficiency is well documented, and the indications are constantly growing, although the impact often is rather limited, as we argue in this review. Bevacizumab is widely used for patients with CRC, while TKI primarily have been used in other solid tumors. Recent evidence suggests that inhibition of angiogenesis may be clinically meaningful through several lines of treatment but lack of biomarkers limits an individualized approach. The tumor microenvironment is anti-immune and a combination of anti-angiogenic drugs and immunotherapy has demonstrated impressive results and may alter the therapy in the years to come.

A significant difference, in terms of standard clinical efficiency parameters, from adding anti-angiogenetic treatments to the existing chemotherapy regimens, have been documented for patients with mCRC. To what extent these differences represent a clinical meaningful benefit is less clear. The process of angiogenesis provides a significant attribution to tumor growth for a fraction of the patients, but unfortunately, it is not identifiable through a single molecular characteristic. This lack of patient selection currently represents the biggest challenge in the field of anti-angiogenetic therapy. Despite this long-lasting challenge, targeting angiogenesis may constitute one of the most important avenues in modern oncology, even after 15 years on the road. Several scenarios contribute to this optimism.

Research within the field of biomarker discovery hasn’t been more intense than now. The introduction of the consensus molecular subtypes, combined with entities such as improved imaging, digital pathology, in vitro testing of tumor biopsies may help to narrow down the field of candidates for whom anti-angiogenetic therapy is crucial for tumor control.

The introduction of new classes of therapy, with anti-angiogenetic properties, may provide additional benefit. Drugs targeting additional angiogenetic factors exemplify this. One example is the monoclonal antibody parsatuzumab (anti–EGFL7) where clinical testing in phase III was halted due to lack of biomarkers [92,93]. Another example may be the modulation of angiomiRs (microRNAS involved in the regulation of angiogenesis) that represent a rather new avenue. This can constitute a mimicking function that compensates for downregulated miRNAs with a tumor-suppressor function, or anti-miRNAs that target elevated oncogenic miRNAs. The first trial results were presented three years ago, with promising results, but this class of therapeutics still face challenges with specific distributions. 

The combination with more natural substances such as vitamin derivatives represents another scenario where the true benefit from these drugs may be revealed even further. Specifically, several pre-clinical studies [94,95,96] have argued for synergy by combining anti-angiogenetics with vitamin–E derivatives (tocotrienol) and clinical documentation have been provided in other cancer types [97]. Results are currently awaited within the field of mCRC.

The biggest potential, however, lies in the combination with immunotherapies, as highlighted in a previous section of this review and the current results and the number of ongoing clinical trials serve as documentation for this standpoint. The combination of these two classes of therapeutics may represent the key to unlocking immunotherapy for the large group of patients with microsatellite stable tumors that currently do not derive benefit from immunotherapy-only strategies. The near future will tell if this forecast holds true. 

## Figures and Tables

**Table 1 cancers-13-01031-t001:** Principal trials comparing first line combination chemotherapy with or without bevacizumab in unselected patients with metastatic colorectal cancer (mCRC).

Author, Year [Ref.] Trial Name (Subgroup)	Regimen	N	RR (%)	*p*	PFS (mo)	HR (95%CI)	OS (mo)	HR (95%CI)
First line								
Kabbinavar, 2003 [6] AVF0780g	5FU	36	17	*p* = 0.03 *p* = 0.43	5.2	0.46 Significant 0.66 NS	13.8	0.63 Significant 1.17 NS
	5FU + bevacizumab 5 mg/kg	35	40		9.0		21.5	
	5FU + bevacizumab 10 mg/kg	33	24		7.2		16.1	
Hurwitz, 2004 [7] AVF2107g	IFL + placebo	411	35	*p* = 0.004	6.2	0.54 Significant	15.6	0.66 Significant
	IFL + bevacizumab	402	45		10.6		20.3	
Kabbinavar, 2005 [8] AVF2192g	5FU + placebo	105	15	*p* = 0.06	5.5	0.50 0.34–0.73	12.9	0.79 0.56–1.10
	5FU + bevacizumab	104	26		9.2		16.6	
Guan, 2011 [9] ARTIST	IFL	64	17	*p* = 0.01	4.2	0.44 0.31–0.63	13.4	0.62 0.41–0.95
	IFL + bevacizumab	139	35		8.3		18.7	
Stathopoulos, 2010 [10]	FLIRI	108	35	NS	–	NS	25.0	NS
	FLIRI + bevacizumab	114	37		–		20.0	
Cunningham, 2013 [11] AVEX	Capecitabine	140	10	*p* = 0.04	5.1	0·53 0.41–0.69	16.8	0·79 0.57–1.09
	Capecitabine + bevacizumab	140	19		9.1		20.7	
Tebbutt, 2010 [12] MAX	Capecitabine	156	30	*p* = 0.16 *p* = 0.006	5.7	0.63 0.50–0.79 0.59 0.47–0.75	18.9	0.88 0.68–1.13 0.94 0.73–1.21
	Capecitabine + bevacizumab	157	38		8.5		18.9	
	Capecitabine + MMC + bevacizumab	158	46		8.4		16.4	
Saltz, 2008 [13] NO 16966	FOLFOX/CapOx + placebo	701	38	NS	8.0	0·83 0.72–0.93	19.9	0·89 0.76–1.03
	FOLFOX/CapOx + bevacizumab	699	38		9.2		21.3	
Passardi, 2015 [14] ITACa	FOLFOX/FOLFIRI	194	50	NS	8.4	0.86 0.70–1.07	21.3	1.13 0.89–1.43
	FOLFOX/FOLFIRI + bevacizumab	176	51		9.6		20.8	

Abbreviations: RR = response rate, PFS = progression free survival, HR = hazard ratio, OS = overall survival, CI = confidence interval, IFL = weekly bolus regimen with irinotecan, 5FU, and leucovorin; FLIRI = bolus regimen with irinotecan, 5FU, and leucovorin adminiatered every three weeks.

**Table 2 cancers-13-01031-t002:** Principal randomized trials comparing second or later line chemotherapy with anti-angiogenic therapy in unselected patients with mCRC.

Author, Year [Ref.] Trial Name (Subgroup)	Regimen	N	RR (%)	*p*	PFS (mo)	HR (95%CI)	OS (mo)	Δ OS HR
2nd line – no prior bevacizumab								
Giantonio, 2007 [26] E3200	FOLFOX	286	9	*p* < 0.0001	4.7	0.61 Significant	10.8	0.75 Significant
	FOLFOX + bevacizumab_HD_	291	23		7.3		12.9	
	* Bevacizumab_HD_	243	3		2.7		10.2	
Peeters, 2013 [27] NCT00752570	FOLFIRI + placebo	49	0	NR	5.2	1.23 0.81–1.86	8.8	0.90 0.53–1.54
	FOLFIRI + trebananib^C^	95	14		3.5		11.9	
2nd line – prior bevacizumab								
Van Cutsem, 2012 [28] VELOUR^A^	FOLFIRI + placebo	614	11	*p* < 0.001	4.7	0.76 0.66–0.87	12.1	0.82 0.71– 0.94
	FOLFIRI + aflibercept	612	20		6.9		13.5	
Bennouna, 2013 [29] ML18147	Chemo	411	4	NS	4.1	0.68 0.59–0.78	9.8	0.81 0.69–0.94
	Chemo + bevacizumab	409	5		5.7		11.2	
Masi, 2015 [30] BEBYP^B^	Chemo	92	17	*p* = 0.57	5.0	0.70 0.52–0.95	15.5	0.77 0.56–1.06
	Chemo + bevacizumab	92	21		6.8		14.1	
Tabernero, 2015 [31] RAISE	FOLFIRI + placebo	536	13	*p* = 0.63	4.5	0.79 0.70–0.90	11.7	0.84 0.73–0.98
	FOLFIRI + ramucirumab	536	13		5.7		13.3	
Hecht, 2015 [32] SPIRITT (KRASwt)	FOLFIRI + bevacizumab	91	19	NR	7.7	1.01 0.68–1.50	18.0	1.06 0.75–1.49
	FOLFIRI + panitumumab	91	32		9.2		21.4	
Cohn, 2013 [33] QUILT–2.018 (KRASmut)	FOLFIRI + placebo	52	2	NR	4.6	1.01 0.61–1.66 0.69 0.41–1.14	12.0	1.27 0.76–2.13 0.89 0.54–1.89
	FOLFIRI + ganitumab	52	8		4.5		12.4	
	FOLFIRI + conatumumab	51	14		6.5		12.3	
3rd line									
Pfeiffer, 2020 [34] EudraCT, 2016–005241–23	TAS–102	47	0	NS	2.6	0.45 0.29–0.72	6.7	0.55 0.32–0.94
	TAS–102 + bevacizumab	46	2		4.6		9.4	

Abbreviations: RR = response rate, PFS = progression free survival, HR = hazard ratio, OS = overall survival, CI = confidence interval, FOLFOX = 5-FU + oxaliplatin, FOLFIRI = 5-FU + irinotecan. NR = not reported, NS = non-significant.

**Table 3 cancers-13-01031-t003:** Principal randomized trials comparing first line combination chemotherapy with single or double targeted therapy.

Author, Year [Ref.] Trial Name (Subgroup)	Regimen	N	RR (%)	OR	PFS (mo)	HR (95%CI)	OS (mo)	Δ OS HR
1st line								
Hecht, 2009 [36] PACCE	FOLFOX + bevacizumab	410	48	0.92 0.70–1.22	11.4	1.27 1.06–1.52	24.5	1.43 1.11–1.83
	FOLFOX + bevacizumab + panitumumab	413	46		10.0		19.4	
Hecht, JCO 2009 PACCE	FOLFIRI + bevacizumab	115	40	1.11 0.65–1.90	11.7	1.19 0.79–1.79	20.5	1.42 0.77–2.62
	FOLFIRI + bevacizumab + panitumumab	115	43		10.1		20.7	
Hecht, JCO 2009 PACCE (RASwt)	FOLFOX + bevacizumab	203	56	–	11.5	1.36 1.04–1.77	24.5	1.89 1.30–2.75
	FOLFOX + bevacizumab + panitumumab	201	50		9.8		20.7	
Tol, 2009 [37] CAIRO2	CapOx + bevacizumab	366	50	*p* = 0.49	10.7	1.22 1.04–1.43	20.3	1.15 NS
	CapOx + bevacizumab + cetuximab	368	53		9.4		19.4	
Tol, 2009 CAIRO2 (RASwt)	CapOx + bevacizumab	156	50	*p* = 0.06	10.6	NS	22.4	NS
	CapOx + bevacizumab + cetuximab	158	61		10.5		21.8	
Saltz, 2012 [38] NCT00252564	FOLFOX + bevacizumab	124	52	NR	11.0	NS	21.3	NS
	FOLF + bevacizumab + cetuximab	123	41		8.3		19.5	
Saltz, 2012 NCT00252564 (RASwt)	FOLFOX + bevacizumab	49	–	NR	10.9	NS	18.8	NS
	FOLF + bevacizumab + cetuximab	46	–		8.8		21.3	
Berlin, 2013 [39] NCT00636610	Chemo + bevacizumab + placebo	101	51	NR	9.3	1.25 0.89–1.76	–	NR
	Chemo + bevacizumab + vismodegib	98	46		10.1		–	
Infante, 2013 [40] NCT00460603	FOLFOX + bevacizumab	43	49	NS	15.9	1.08 0.47–2.45 1.49 0.75–2.98	21.6	1.16 0.66–2.03 0.94 0.54–1.65
	FOLFOX + axitinib	42	29		11.0		18.1	
	FOLFOX + axitinib + bevacizumab	41	39		12.5		19.7	

Abbreviations: RR = response rate, OR = odds ratio, PFS = progression free survival, HR = hazard ratio, OS = overall survival, CI = confidence interval, FOLFOX = 5-FU + oxaliplatin, FOLFIRI = 5-FU + irinotecan, CapOx = capecitabine + oxaliplatin, FOLF = F-FU and lecovorin. Vismodegib: Hedgehog pathway inhibitor Abbreviations: NS = non-significant, NR = not reported.

**Table 4 cancers-13-01031-t004:** Principal randomized trials comparing chemotherapy with or without tyrosine kinase inhibitor (TKI) as first or second line therapy in unselected patients with mCRC.

Author, Year [Ref.] Trial Name (Subgroup)	Regimen	N	RR (%)	*p*	PFS (mo)	HR (95%CI)	OS (mo)	HR (95%CI)
First line								
Hecht, 2011 [45] CONFIRM I	FOLFOX + placebo	583	46	*p* > 0.05	7.6	0.88 0.74–1.03	20.5	1.08 0.94–1.26
	FOLFOX + vatalanib	585	42		7.7		21.4	
Hoff, 2012 [46] HORIZON II	FOLFOX/CapOx + placebo	358	50	*p* = 0.90	8.3	0.84 0.73–0.94	18.9	0.94 0.79–1.13
	FOLFOX/CapOx + cediranib	502	51		8.6		19.7	
	FOLFOX/CapOx + cediranib	216	Terminated, 20 mg sufficient					
Schmoll, 2012 [47] HORIZON III	FOLFOX + bevacizumab	713	47	*p* = 0.67	10.3	1.10 0.97–1.25	21.3	0.95 0.82–1.10
	FOLFOX + cediranib	709	46		9.9		22.8	
Carrato, 2013 [48] NCT00457691	FOLFIRI + placebo	382	34	*p* = 0.68	8.4	1.10 0.89–1.34	19.8	1.17 0.94–1.47
	FOLFIRI + sunitinib	386	32		7.8		20.3	
Hecht, 2015 [49] NCT00609622	FOLFOX + bevacizumab	95	40	*p* = 0.70	15.4	2.37 1.15–4.85	34.1	1.47 0.91–2.3
	FOLFOX + sunitinib	96	43		9.3		23.7	
Tabernero, 2013 [50] RESPECT	FOLFOX + placebo	101	59	NR	8.7	0.88 0.64–1.23	18.1	1.13 0.79–1.61
	FOLFOX + sorafenib	97	44		9.1		17.6	
Second line								
Van Cutsem, 2011 [51] CONFIRM II	FOLFOX + placebo	429	–	NS	4.2	0.63 0.48–0.83	11.9	0.82 0.63–1.06
	FOLFOX + vatalanib	426	–		5.6		13.1	
Cunningham, 2012 [52] HORIZON I	FOLFOX + bevacizumab	66	27	NS	7.8	1.28 0.85–1.95 1.17 0.77–1.76	19.6	1.39 0.92–2.09 1.00 0.66–1.50
	FOLFOX + cediranib 20	71	18		5.8		14.3	
	FOLFOX + cediranib 30	73	20		7.2		16.8	
Bendell, 2013 [53] NCT00615056	FOLFIRI + bevacizumab	51	24	NR	6.9	1.27 0.77–2.11	15.7	1.36 0.82–2.24
	FOLFIRI + axitinib	49	24		5.7		12.9	
	FOLFOX + bevacizumab	35	20		6.4	1.04 0.55–1.96	14.1	0.69 0.37–1.27
	FOLFOX + axitinib	36	19		7.6		17.1	
Sanoff, 2018 [54] NCT01298570	FOLFIRI + placebo	61	20	*p* = 0.21	5.3	0.73 0.53–1.01	11.7	1.01 0.71–1.44
	FOLFIRI + regorafenib	120	29		6.1		13.8	

Abbreviations: RR = response rate, PFS = progression free survival, HR = hazard ratio, OS = overall survival, CI = confidence interval, FOLFOX = 5-FU + oxaliplatin, FOLFIRI = 5-FU + irinotecan, CapOx = capecitabine + oxaliplatin, NS = non-significant, NR = not reported.

**Table 5 cancers-13-01031-t005:** Principal randomized trials in which TKIs were used in third or later line therapy in patients with mCRC.

Author, Year [Ref.] Trial	Regimen	N	RR (%)	*p*	PFS (mo)	HR (95%CI)	OS (mo)	Δ OS HR
Third or later line								
Grothey, 2013 [55] CORRECT	Placebo	255	0	0.19	1.7	0.49 0.42–0.58	5.0	0.77 0.64–0.94
	Regorafenib	505	1		1.9		6.4	
Li, 2014 [56] CONCUR	Placebo	68	0	0.045	1.7	0.31 0.22–0.44	6.3	0.55 0.40–0.77
	Regorafenib	136	4		3.2		8.8	
Bekaii–Saab, 2019 [57] ReDOS	Regorafenib 160	62	–		2.0	0.84 0.57–1.24	6.0	0.72 0.47–1.10
	Regorafenib 80 ⇨ 160 mg	54	–		2.8		9.8	
Argiles, 2019 [58] REARRANGE	Regorafenib 160	101	–		1.9	NS	7.4	NS
	Regorafenib 120 ⇨ 160 mg	99	–		2.0		8.6	
	Regorafenib 160 1w	99	–		2.0		7.1	
Siu, 2013 [59] AGITG CO.20 (KRASwt)	Cetuximab	374	7	0.004	3.4	0.72 0.62–0.84	8.1	0.88 0.74–1.03
	Cetuximab + brivanib	376	14		5.0		8.8	
Li, 2013 [60] FRESCO	Placebo	138	0	0.01	1.8	0.26 0.21–0.34	6.6	0.65 0.51–0.83
	Fruquintinib	278	5		3.7		9.3	
Van Cutsem, 2018 [61] LUME	Placebo	382	0	>0.05	1.4	0.58 0.49–0.69	6.0	1.01 0.86–1.19
	Nintedanib	386	0		1.5		6.4	
Samalin, 2020 [62] PRODIGE27 (KRASmut)	Irinotecan	57	2		1.9	2.37 0.97–1.25	6.3	1.47 0.91–2.3
	Sorafenib	57	2		2.1		5.6	
	Irinotecan + Sorafenib	59	4		3.6		7.2	
Eng, 2019 [63] IMBlaze	Regorafenib	90	2	NS	2.0	1.39 ^A^ 1.00–1.94 1.25 ^B^ 0.94–1.65	8.5	1.19 ^A^ 0.83–1.71 1.00 ^B^ 0.73–1.38
	Atezolizumab	90	2		1.9		7.1	
	Atezolizumab + cobimetinib	183	5		1.9		8.9	

Abbreviations: RR = response rate, PFS = progression free survival, HR = hazard ratio, OS = overall survival, CI = confidence interval, NS = non significant, (A) regorafenib vs. atezolizumab, (B) regorafenib vs. cobimetinib.

**Table 6 cancers-13-01031-t006:** Ongoing clinical trials in metastatic colorectal cancer combining immunotherapy and anti-angiogenetic treatment.

Trial ID	Phase	Immunotherapy	Anti-Angiogenetic	Treatment	Status
NCT 04362839.	I	Ipilimumab + Nivolumab	Regorafenib	Last line	Recruiting
NCT 04110093	I/II	PD–1 inhibitor ^1^	Regorafenib	Last line	Recruiting
NCT 03647839	II	Nivolumab	BNC105	Last line	Recruiting
NCT 04030260	II	Nivolumab	Regorafenib	Last line	Recruiting
NCT 04126733	II	Nivolumab	Regorafenib	Last line	Not recruiting
NCT 03712943	I	Nivolumab	Regorafenib	Last line	Recruiting
NCT 03239145	I	Pembrolizumab	Trebananib	Last line	Recruiting
NCT 03797326	II	Pembrolizumab	Lenvatinib	Last line	Recruiting
NCT 03396926	II	Pembrolizumab	Bevacizumab	Last line	Recruiting
NCT 03657641	I/II	Pembrolizumab	Regorafenib	Last line	Recruiting
NCT 03170960	I/II	Atezolizumab	Cabozantinib	Last line	Recruiting
NCT 02873195	II	Atezolizumab	Bevacizumab	Last line	Not recruiting
NCT 02997228	III	Atezolizumab	Bevacizumab	First line	Suspended
NCT 03721653	II	Atezolizumab	Bevacizumab	First line	Recruiting
NCT 02982694	II	Atezolizumab	Bevacizumab	Last line	Recruiting
NCT 03475953	I/II	Avelumab	Regorafenib	Second line	Recruiting
NCT 03050814	II	Avelumab	Bevacizumab	First line	Not recruiting
NCT 03376659	I/II	Durvalumab	Bevacizumab	Second line	Recruiting
NCT 03539822	I	Durvalumab	Cabozantinib	Last line	Recruiting
NCT 03851614	II	Durvalumab	Cediranib	Last line	Recruiting
NCT 02484404	I/II	Durvalumab	Cediranib	Last line	Recruiting

^1^ Of investigators choice; Atezolizumab (anti-PD–L1); avelumab (anti-PD–L1); bevacizumab (anti-vascular endothelial growth factor A); BNC105 (a vascular disrupting agent); cabozantinib (tyrosine kinase inhibitor of c–MET and vascular endothelial growth factor receptor 2); cediranib (tyrosine kinase inhibitor of vascular endothelial growth factor receptors 1–3); durvalumab (anti-PD–1); ipilimumab (anti-CTLA–4); lenvatinib (tyrosine kinase inhibitor of vascular endothelial growth factor receptors 1–3); nivolumab (anti-PD–1); pembrolizumab (anti-PD–1); regorafenib (tyrosine kinase inhibitor of TIE2 and vascular endothelial growth factor 2); trebananib (anti-angiopoietin–2).

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
