# Peer review of "Angiogenesis Inhibitors for Colorectal Cancer. A Review of the Clinical Data"

_cancers, 2021, doi:10.3390/cancers13051031_

Round 1
Reviewer 1 Report
Authors have addressed my comments and the paper was improved.
Reviewer 2 Report
The authors have made the revision asked for and I find that the paper is strongly improved
This manuscript is a resubmission of an earlier submission. The following is a list of the peer review reports and author responses from that submission.
Round 1
Reviewer 1 Report
Angiogenesis inhibitors for colorectal cancer by Torben Frøstrup Hansen , Camilla Qvortrup and Per Pfeiffer.
This review is very descriptive without raising the fundamental problem of why anti-angiogenic therapies are expected to fail unless you are targeting cancer specific blood vessel markers without affecting the surrounding normal blood vessels. What is different about cancer blood vessels that enable you to selectively target them? Is this local or systemic delivery?
I do not understand what the reader gets from this review apart from re-enforcing the well-established role of VEGF and to some extent EGFR in angiogenesis. Reading this review should put people off systemic anti-angiogenic therapy. The use of poor English does not help it either.
Reviewer 2 Report
This review highlights the current status of angiogenesis inhibitors in colorectal cancer. It focuses on clinical papers and clinical trial reports that addressed anti-angiogenic drugs in combination with chemotherapy and/or checkpoint inhibitors. Authors concluded that combination of anti-angiogenics with immunotherapy might provide efficient response as based on some preliminary reports form abstracts presented in congresses and few published reports of small clinical trials. They provided the current list of ongoing clinical trials of angiogenesis inhibitors in combination with immunotherapy, mostly as second or last line therapy.
Overall, the paper is well written and is of high interest for clinician and scientists who are searching for biomarkers of CRC response to the combination of angiogenesis inhibitors and immunotherapy.
Some precisions are required for improvement.
- Line 54; please precise whether RAS mutation status affects the response of CRC to antiangiogenics ?
- Line 173; referring to the following statement “Unfortunately, in two large randomized trials, no gain was measured in terms of OS in patients with CRC, and in one study, there was actually fewer patients alive after ten years than in the control group [41,42] », please discuss the effect of reduced OS in patients treated with angiogenesis inhibitors reported in these two references and whether this is a consequence of side effects or a relapse after therapy cessation?
- Line 308; if the elevated IFP hampers the entry of TILs in tumors, it should also block the entry of Treg. Authors might clarify how leaky blood vessels can generate an immunosuppressive TME (REF 87 and 79) ?.
- Line 254; resistance mechanisms were not described, while intensive preclinical investigations have been reported in the last decade in some high IF journals. Authors might discuss briefly some key resistance mechanisms in this section.
- Line 409; please add a reference on parsatuzumab if the study was published
- Line 417; please add references of the preclinical studies that argued for synergy by combining anti-angiogenesis inhibitors with Vit-E (tocotrienol).
Reviewer 3 Report
The review-paper deals with the very important topic of anti-angiogenesis in colon cancer.
However I find it rather superficial, with many relevant aspects not discussed.
Major comments
a) the difference between left and right colonic cancer is not exhamined, neither the basis to choose anti-angiogenic versus anti-EGFr receptor antagonist:
Left colonic RAS wild-type metastatic colorectal cancers derive major clinical benefit from anti-EGFR versus bevacizumab therapy, while bevacizumab appears to have clear advantage in RAS wild-type metastatic right-sided tumors.
b) cardiovascular side effects of anti-angiogenic therapy are only briefly mentioned, while a more accurate description is necessary, since induction of hypertension and consequences are known since a long time
c) the discussion on the rationale beyond combination of immuneoherapy and anti-angiogenic therapy is not deep enough
d) Aflibercept (AFL) or VEGF-Trap is not an monoclonal antibody, but a decoy receptor, actually is an Fc fusion protein, the description in the paper is confusing
e) I find the review more a list of drugs, without much insight on what would be the right clinical choice
d) finally: metastasis and how anti-angiogenic therapy can acts on it is not discussed. Bevacizumab was registred for use in metastatic CRC after the paper of Hurwitz, 2004 (cited in the text), so plenty of data are available.
English is very poor with several mistakes, in particular the "simple summary" should be toataly re-wrtitten; for instance "have" is used totally randomly after a single subject, but it is plural! (example: The introduction of immunotherapy HAVE opened)